# Value of the Neutrophil to Lymphocyte Ratio and Platelet to Lymphocyte Ratio in Predicting CPET Performance in Patients with Stable CAD and Recent Elective PCI

**DOI:** 10.3390/medicina58060814

**Published:** 2022-06-16

**Authors:** Andrei Drugescu, Mihai Roca, Ioana Mădălina Zota, Alexandru-Dan Costache, Oana Irina Gavril, Radu Sebastian Gavril, Teodor Flaviu Vasilcu, Ovidiu Mitu, Irina Mihaela Esanu, Iulia-Cristina Roca, Cristina Mihaela Ghiciuc, Florin Mitu

**Affiliations:** 1Medical I Department, Faculty of Medicine, “Grigore T. Popa” University of Medicine and Pharmacy, 700115 Iasi, Romania; andreidrugescu@yahoo.com (A.D.); adcostache@yahoo.com (A.-D.C.); ir.ungureanu@yahoo.com (O.I.G.); rgavril87@yahoo.com (R.S.G.); teodor.vasilcu@gmail.com (T.F.V.); mituovidiu@yahoo.co.uk (O.M.); irina.esanu@umfiasi.ro (I.M.E.); mitu.florin@yahoo.com (F.M.); 2Surgery II Department, Faculty of Medicine, ”Grigore T. Popa” University of Medicine and Pharmacy, 700115 Iasi, Romania; iuliaroca@yahoo.com; 3Morpho-Functional Sciences II Department, Faculty of Medicine, “Grigore T. Popa” University of Medicine and Pharmacy, 700115 Iasi, Romania; cristina.ghiciuc@umfiasi.ro

**Keywords:** functional capacity, platelet to lymphocyte ratio, neutrophil to lymphocyte ratio, cardiopulmonary exercise test, oxygen uptake, coronary artery disease

## Abstract

*Background and Objectives*: Functional capacity (FC) assessed via cardiopulmonary exercise testing (CPET) is a novel, independent prognostic marker for patients with coronary artery disease (CAD). Neutrophil to lymphocyte ratio (NLR) and platelet to lymphocyte ratio (PLR) are two readily available predictors of systemic inflammation and cardiovascular event risk, which could be used as cost-effective predictors of poor FC. The purpose of this study was to evaluate the utility of NLR and PLR in predicting poor FC in patients with CAD and recent elective percutaneous coronary intervention (PCI). *Materials and Methods*: Our cross-sectional retrospective analysis included 80 patients with stable CAD and recent elective PCI (mean age 55.51 ± 11.83 years, 71.3% male) who were referred to a cardiovascular rehabilitation center from January 2020 to June 2021. All patients underwent clinical examination, cardiopulmonary exercise testing on a cycle ergometer, transthoracic echocardiography and standard blood analysis. *Results*: Patients were classified according to percent predicted oxygen uptake (% VO2 max) in two groups—poor FC (≤70%, *n* = 35) and preserved FC (>70%, *n* = 45). There was no significant difference between groups regarding age, gender ratio, presence of associated comorbidities, left ventricular ejection fraction and NLR. PLR was higher in patients with poor FC (169.8 ± 59.3 vs. 137.4 ± 35.9, *p* = 0.003). A PLR cut-off point of 139 had 74% sensitivity and 60% specificity in predicting poor FC. After multivariate analysis, PLR remained a significant predictor of poor functional status. *Conclusions*: Although CPET is the gold standard test for assessing FC prior to cardiovascular rehabilitation, its availability remains limited. PLR, a cheap and simple test, could predict poor FC in patients with stable CAD and recent elective PCI and help prioritize referral for cardiovascular rehabilitation in high-risk patients.

## 1. Introduction

Coronary artery disease (CAD) is a significant public health problem, with a substantial contribution to global morbidity and mortality, especially in low- and middle-income countries [1,2]. Current guidelines firmly recommend enrollment in a comprehensive cardiovascular rehabilitation (CR) program after CAD diagnosis or revascularization, with proven beneficial effects on cardiovascular and all-cause mortality and individual quality of life (QoL) [3,4,5,6]. CAD is associated with a significant impact on the individual’s exercise capacity, which can rapidly deteriorate after a major cardiovascular event [7]. Compared to post-acute coronary syndromes (ACS), CR addressability is lower in patients with stable CAD and following elective percutaneous coronary intervention (PCI) [8], as well as in women, elderly and socio-economically deprived patients [9].

Functional capacity (FC) is a strong, independent prognostic factor in heart failure (HF) [10] and CAD [11]. The prognostic value of FC is independent and additive to other well established mortality predictors such as left ventricular ejection fraction (LVEF), smoking, hypertension (HTN), dyslipidemia and diabetes [12,13,14]. Peak oxygen uptake (VO2 max) assessed via cardiopulmonary exercise testing (CPET) is an objective measure of FC, and an independent predictor of cardiovascular morbidity and mortality in patients with CAD [12,13,15,16]. However, CPET availability remains limited, especially in developing countries.

Systemic inflammation plays a major role in CAD etiopathogenesis [17] and routine inflammatory biomarkers (complete blood count, C-reactive protein) have proven their role for both acute and long-term cardiovascular risk assessment [18,19,20]. Physical activity decreases systemic markers of inflammation, thrombosis and endothelial dysfunction, and has a key role in preventing CAD [21,22,23]. The platelet to lymphocyte ratio (PLR) is an integrated reflection of two important opposite inflammatory pathways that can be easily calculated from a complete blood count. PLR initially served as a prognostic biomarker in neoplastic diseases [24,25], but has recently been studied in HF [26,27,28], ACS [29,30,31,32,33], atrial fibrillation [34,35], deep venous thrombosis [36], PCI [37,38,39] and infective endocarditis [40]. The neutrophil to lymphocyte ratio (NLR) is another readily available biomarker of inflammation in cardiac and non-cardiovascular disorders [41,42,43]. In previous reports, the NLR appeared to be a predictor of cardiovascular events and mortality in patients with stable CAD and was associated with coronary atherosclerosis severity [44,45]. NLR was also used as a predictor for functional capacity in patients undergoing CR [46] and a predictor of lipid-lowering effectiveness in patients with familial hypercholesterolemia and atherosclerotic cardiovascular disease [47]. However, the current literature offers limited data regarding the role of these readily available inflammatory biomarkers in predicting exercise performance in CAD patients. We therefore hypothesized that impaired cardiovascular performance (as defined by CPET) could be predicted by NLR and PLR in individuals with stable CAD and recent elective PCI. The aim of this study was to evaluate the utility of two readily available inflammatory biomarkers (NLR and PLR) in predicting poor FC in patients with CAD and recent elective PCI.

## 2. Materials and Methods

We conducted a retrospective cross-sectional study of all patients with stable CAD and recent elective PCI, referred for phase II CR in the Cardiovascular Unit of the Clinical Rehabilitation Hospital in Iași over a period of 18 months (January 2020–June 2021). The Cardiovascular Unit of the Clinical Rehabilitation Hospital in Iași is a nationally ranked dedicated rehabilitation center specializing in phases II and III of cardiovascular rehabilitation [4,48]. Inclusion criteria were as follows: elective PCI performed for stable CAD during the previous 3 months and CPET performed upon admission (Figure 1). Patients with ACSduring the previous 12 months, anemia (hemoglobin <12 g/dL in females and <13 g/dL in males), atrial fibrillation, moderate or severe valvular heart disease, decompensated congestive heart failure, any congenital heart disease or any other severe chronic disease except CAD were excluded from this analysis. All patients had a negative COVID-19 PCR upon admission. Socio-demographic, clinical, biological, CPET and echocardiographic data were extracted from hospital medical records.

All patients were under optimal CAD treatment, according to current guidelines [49]. Obesity was defined as a body mass index (BMI) ≥30 kg/m^2^. High blood pressure (HBP) was defined as current BP lowering treatment, prior diagnosis of HBP, resting systolic blood pressure (SBP) greater than 140 or resting diastolic blood pressure (DBP) greater than 90 mmHg [50]. Diabetes was defined as current antidiabetic treatment, previous diabetes diagnosis, fasting glucose ≥126 mg/dL on two separate occasions or a value for glycosylated hemoglobin ≥6.5% [51,52,53].

According to hospital protocol, blood samples were collected a jeun, in the morning upon admission, by qualified medical professionals. All blood samples were processed in the hospital’s laboratory. Complete blood count was processed using the Pentra DF Nexus Hematology System^®^ (Horiba Healthcare, Kyoto, Japan). Biochemistry was processed using the Transasia XL 1000 Fully Automated Biochemistry Analyzer (Transasia Bio-Medicals Ltd., Mumbai, India). We recorded the following parameters: platelet count, neutrophil count, lymphocyte count, C-reactive protein (CRP), low-density lipoprotein (LDL) and glycated hemoglobin (HbA1c). NLR was calculated using the absolute neutrophil (N) and lymphocyte (L) values from the complete blood count, using the formula: NLR = N/L. PLR was calculated using the absolute platelets (P) and lymphocyte (L) values from the complete blood count, using the formula: PLR = P/L. 

Standardized transthoracic echocardiography (2D, Doppler) was performed by experienced sonographists according to current EACVI guidelines [54] (Toshiba Aplio 500 Series, Toshiba Medical Systems Corporation, Ōtawara, Tochigi, Japan) prior to CPET evaluation. LVEF was calculated using Simpson biplane method.

CPET was performed by a certified pulmonologist on the Piston PRE-201 ergospirometer (Piston Ltd., Budapest, Hungary). According to hospital protocol, CPET was performed in the morning of the second day of hospitalization, in order to establish functional capacity and target heart rate for exercise rehabilitation. Each patient signed a written informed consent before the test. The test consisted of a 2 min resting period followed by 3 min warm up at 0 W followed by standard incremental exercise protocol of 15 W/min. The CPET was performed under continuous heart rate (HR), 12-lead ECG (electrocardiographic) and pulse oximetry (SpO2) monitoring. Blood pressure was recorded every 2 min. Indications for exercise termination included exhaustion, myocardial ischemia, complex ventricular arrythmia, grade 2 or 3 atrio-ventricular block, a sudden drop in BP levels > 20 mmHg, extreme BP elevation (SBP > 220 mmHg, DBP > 120 mmHg), SpO2 < 80%, confusion or severe dizziness. We recorded the following parameters: resting SBP and DBP (measured with a manual sphygmomanometer immediately prior to the CPET), resting HR (recorded on the resting ECG performed immediately prior to the CPET), % peak HR (maximum heart rate relative to predicted normal for age (220—age in years)), % peak WR (maximum workload relative to predicted normal according to age and sex, automatically calculated by the ergospirometer software) and % VO2 max (maximum oxygen uptake (highest value, mean of 20 s) relative to predicted normal according to age and sex, automatically calculated by the ergospirometer software). Functional capacity was assessed according % VO2 max, using a convention proposed by Cooper et al., as follows: >80%—normal, 71–80%—mildly reduced, 51–70%—moderately reduced and ≤50%—severely reduced [55]. Due to a relatively small number of enrolled patients, we divided our study group as follows: poor FC (% VO2 max ≤70) and preserved FC (% VO2 max >70).

### 2.1. Statistical Analysis

Data analysis was performed using SPSS 20.0 (Statistical Package for the Social Sciences, Chicago, IL, USA). For continuous data, the normality of distribution was assessed by Shapiro–Wilk test. Data are presented as mean ± standard deviation (SD) for continuous variables with normal distribution, or as median with interquartile range for non-normally distributed continuous variables. Categorical variables are presented as number of cases with percent frequency. An independent samples T-test was used to compare continuous variables with normal distribution. A non-parametric Mann–Whitney’s U test was applied to compare the variables not satisfying the assumption of normality. Categorical comparisons were performed using Chi-square test or Fisher’s exact test (when the expected number of values in any of the cells of a contingency table was ≤5). Variables with *p* < 0.05 in the univariate analysis were included in the multivariate logistic regression model, to assess the independent predictors of poor FC (% VO2 max ≤ 70). The results are presented as odds ratio (OR) with 95% confidence intervals (CIs). Receiver operating characteristic (ROC) curve analysis was done to determine the optimum cut-off value of PLR in predicting poor FC of CAD patients and recent PCI. Correlation analyses, calculating Pearson correlation coefficients, were assessed considering normally distributed and linearly related variables. A two-sided *p* value < 0.05 was considered significant for all analyses.

### 2.2. Ethics Statement

The study was approved by the Review Board/Ethics Committee of the Clinical Rehabilitation Hospital Iași (number 28567/21.12.2020) and complied with the Declaration of Helsinki. The Clinical Rehabilitation Hospital Iasi Review Board/Ethics Committee considered informed consent unnecessary owing to the characteristics of this study (retrospective database analysis).

## 3. Results

Table 1 illustrates clinical and demographic features, laboratory findings and exercise measurements of the 80 analyzed patients (age range: 34–79 years old) and a univariate analysis of the two subgroups according to the values of % VO2 max. Age and the presence of cardiometabolic comorbidities (obesity, diabetes, HTN, LDL level) were similar among the two subgroups.

Our analysis included 35 patients with % VO2 ≤70 and 45 patients with % VO2 >70. Among the hematological parameters, the PLR was higher in the group of % VO2 max ≤70 than in the group of % VO2 max >70 (*p* = 0.003, Figure 2). NLR values were higher in patients with poor FC, but the difference did not reach statistical significance. Patients with preserved FC had higher LVEF values (*p* = 0.003) and reached a higher peak HR during exercise (*p* = 0.006). CRP, platelet, neutrophil and lymphocyte count, as well as resting HR and blood pressure values, were similar between the two subgroups.

PLR were positively correlated with % VO2 max (*p* < 0.05; Table 2). NLR was associated with PLR, but not with the analyzed CPET parameters.

In a logistic multivariate model, the PLR remained significant predictor of poor FC (Table 3). NLR was not a significant predictor of poor FC in univariate analysis; thus, it was not included in the multivariable regression model.

ROC curves explored the relationship between the PLR and FC. Using a cut-off point of 139, the PLR predicted poor FC with a sensitivity of 74% and specificity of 60% (ROC area under curve: 0.681, 95% CI: 0.563–0.799, *p* = 0.006; Figure 3).

## 4. Discussion

The results of the present study suggest that significant prognostic information can be obtained from routine blood test results in CAD patients undergoing CR. Walzik et al. recently published reference values for NLR and PLR, encouraging a more frequent use in clinical practice [56]. NLR and PLR were significantly higher among our patients (especially in the subgroup with poor functional capacity) compared to the average NLR and PLR values recently reported in a healthy population-based cohort: 1.76 (0.83–3.92) and 120 (61–239), respectively [57].

Functional capacity is an independent prognostic factor in CAD patients [4,15,58,59,60]. Our data show that impaired FC assessed with CPET is associated with changes in leukocyte subsets and platelets. Inflammation plays a role in the onset, progression and destabilization of atherosclerotic plaque. Systemic inflammation is known to be associated with parietal vascular inflammation [61]. Activation of lymphocytes and monocytes is essential in the early stages of atherosclerosis, while neutrophils are implicated in plaque destabilization and thrombosis [62]. NLR is an easily available biomarker of vascular parietal inflammation [63] with documented prognostic implications in various cardiovascular diseases [64]. Elevated NLR has been associated with an increased risk of atrial [65,66,67] and ventricular arrythmias [68] and with worse outcomes in acute decompensated heart failure [69] and acute coronary syndromes [70]. Besides CAD, NLR offers prognostic information in patients with abdominal aortic aneurysm [71], chronic threatening limb ischemia [72] and other cardiovascular emergencies [73,74,75,76,77]. NLR is also a biomarker of interest in severe mitral and aortic valvular heart disease [78,79] and is a predictor of poor FC in patients with HF (OR 3.085, 95% CI 1.52–6.26, *p* = 0.002) [80]. Indeed, immune dysregulation is known as an important characteristic of poor aerobic capacity. Increased NLR could be associated with poorer physical performance in CAD patients and with lower LVEF in patients with HF [80]. In a previous study, Yıldız et al. showed that a threshold level of 2.26 for NLR predicts a poor FC (sensitivity of 83% and specificity of 69%) in patients with idiopathic dilated cardiomyopathy [81]. In another study of 94 patients with compensated HF, NLR was correlated with exercise performance, and a cut-off point of 2.74 was established for predicting poor FC [80]. FC was expressed as maximal exercise intensity (METs) during treadmill test in both previous studies, a less specific marker for FC compared to % VO2 max. In the present analysis, although NLR values were higher in patients with poor FC, the difference did not reach statistical significance.

Elevated blood and plasma viscosity have been associated with an increased risk of CAD. CAD patients have increased platelet and monocyte aggregates in their bloodstream, which are associated with plaque instability, worse in-hospital outcomes and an increased risk of future cardiac events [82,83]. Exercise training improves blood rheology, which may contribute to the increased FC observed after CR [84]. PLR reflects the balance between thrombotic and inflammatory pathways, being influenced by blood viscosity and inflammation [63,82]. Ayca et al. showed that patients with high PLR had higher Syntax Score (SXS) and a PLR > 137 had a specificity of 52% and a sensitivity of 61% for predicting SXS > 22, marking PLR as a prognostic marker in primary PCI [39]. Azab et al. examined the prognostic value of PLR in non-STEMI [31]. At the 4-year follow up, patients with PLR > 176 had a 42% all-cause mortality, whereas patients with PLR < 118.4 had an all-cause mortality rate of 17%. In another study of patients with STEMI, Ugur et al. found that patients with PLR > 174.9 had higher all-cause mortality at 6 months compared with patients with PLR < 174.9 [32]. Moreover, previous studies also showed that high PLR is associated with increased risks of new-onset atrial fibrillation [38], contrast-induced acute kidney injury [85], more advanced HF [29] and no reflow after PCI in STEMI patients.

Our analysis shows that PLR is higher in patients with % VO2 max ≤70 than in patients with %VO2 max >70 (*p* = 0.003, Figure 2). PLR was positively correlated with % VO2 max (*p* < 0.05; Table 2) and remained significant predictor of poor FC (OR, 1.015; 95% CI, 1.004–1.027; *p* = 0.009) after multivariate analysis. Using a cut-off point of 139, the PLR predicted poor FC with a sensitivity of 74% and specificity of 60% (ROC area under curve: 0.681, 95% CI: 0.563–0.799, *p* = 0.006; Figure 3).

Other studies have assessed the relationship between CRP and FC in various non-cardiovascular conditions [86,87,88]. Our results do not support a significant association between CRP and FC in CAD patients with recent PCI.

The results of the present study suggest that significant prognostic information can be obtained from routine blood test results in CAD patients with recent PCI. Because the PLR is a ratio, it is less prone to bias/variations than individual blood parameters that can be altered by several variables (e.g., dehydration, over-hydration and blood specimen handling). To our knowledge, this is the first study that evaluated whether PLR can predict FC assessed by CPET in stable CAD with recent PCI.

This study has several limitations. Most importantly, this single-center retrospective analysis included a relatively small number of patients and did not include measurement of other important cardiac biomarkers such as troponin and natriuretic peptides. Furthermore, although all patients had a negative COVID-19 PCR upon admission, we were not able to accurately exclude prior COVID-19 infection (we did not perform antibody testing and we did not take into consideration vaccination status). Previous COVID-19 infection can negatively impact functional capacity and could influence our results. The retrospective structure of our study and the small number of cases, our multiple regression was limited to only a few covariates. Residual covariates and additional risk factors (for example smoking status) could significantly impact our results. Considering these limitations, our conclusions need to be validated in larger cohort analyses. Furthermore, larger prospective studies are needed to evaluate whether PLR can also predict FC improvement after cardiovascular rehabilitation programs.

## 5. Conclusions

PLR is higher in patients with recent PCI for stable CAD and poor FC compared to those with preserved FC. FC is an independent predictor of long-term prognosis in CAD. Although CPET is the gold standard test for assessing FC prior to cardiovascular rehabilitation, its availability remains limited. PLR, a cheap and simple test, could predict poor functional capacity in patients with stable CAD and recent elective PCI and help prioritize referral for cardiovascular rehabilitation in high-risk patients.

## Figures and Tables

**Figure 1 medicina-58-00814-f001:**
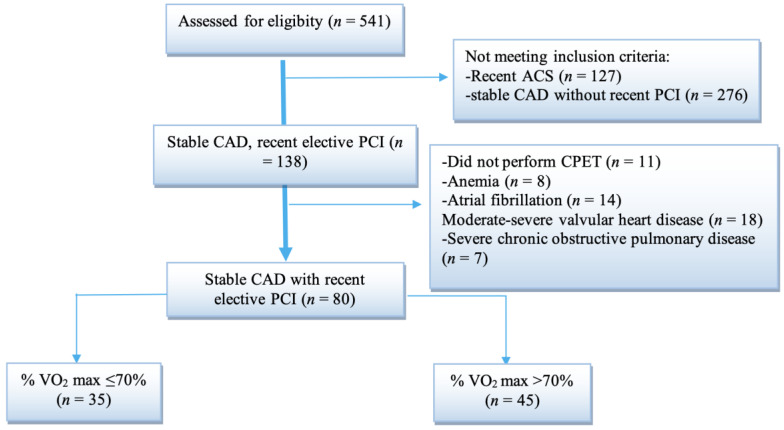
Flow chart diagram of patients hospitalized in the Cardiovascular Rehabilitation Clinic Unit between January 2020 and June 2021. CAD—coronary artery disease, PCI—percutaneous coronary intervention, ACS—acute coronary syndrome, CPET—cardiopulmonary exercise test, % VO2 max—percentage of the predicted value of maximal oxygen uptake.

**Figure 2 medicina-58-00814-f002:**
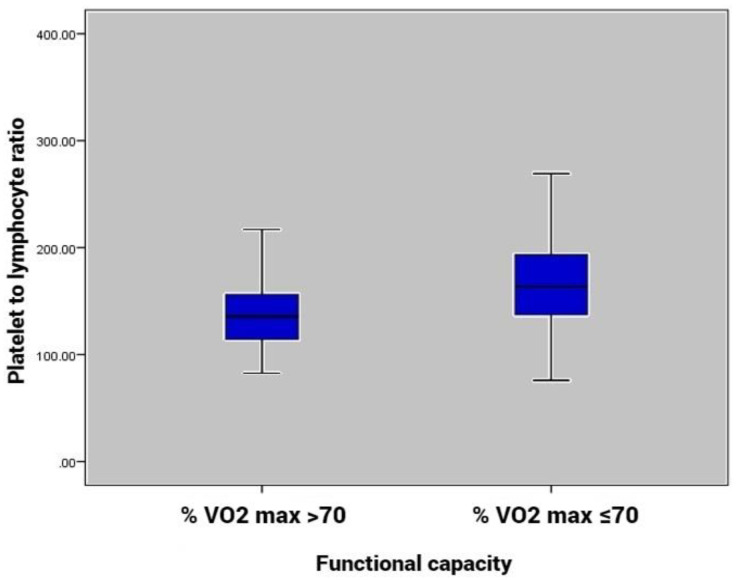
Platelet to lymphocyte ratio levels according to functional capacity groups.

**Figure 3 medicina-58-00814-f003:**
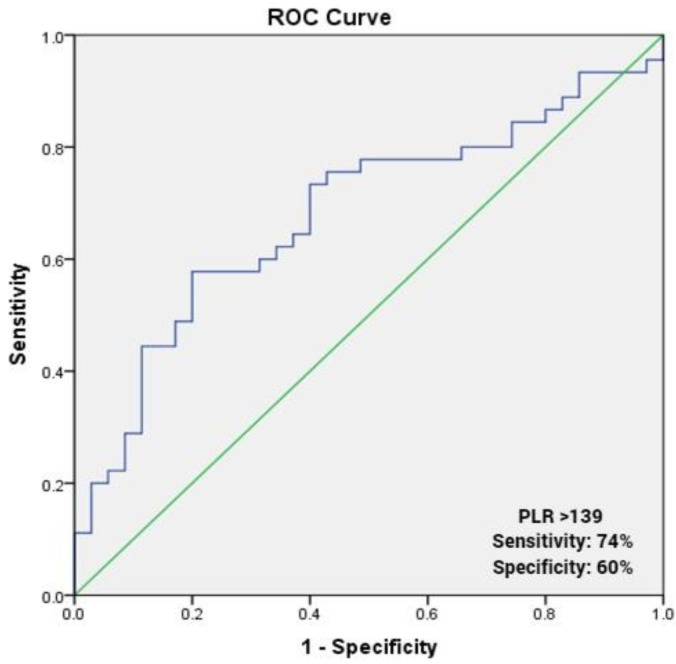
Receiver operating characteristic curve of platelet to lymphocyte ratio for predicting poor functional capacity.

**Table 1 medicina-58-00814-t001:** Univariate analysis of the two groups according to the values of % VO2 max in all study participants.

Parameters	All Patients (*n* = 80)	% VO2 Max >70 (*n* = 45)	% VO2 Max ≤70 (*n* = 35)	*p* Value *
Age (years) ^×^	55.51 ± 11.83	57.02 ± 12.08	53.57 ± 11.38	0.19
NLR ^×^	1.97 ± 0.80	1.83 ± 0.65	2.15 ± 0.93	0.07
PLR ^×^	155.6 ± 52.7	137.4 ± 35.9	169.8 ± 59.3	0.003
Platelet count, ×10^3^/μL ^×^	256 ± 60	244.4 ± 56.1	266.3 ± 56.1	0.11
Neutrophil count, ×10^3^/μL ^×^	3.32 ± 1.25	2.92 ± 0.90	3.83 ± 1.45	0.001
Lymphocyte count, ×10^3^/μL ^†^	1.72 (1.44–1.99)	1.45 (1.31–2.43)	1.86 (1.65–1.88)	0.06
CRP (mg/dl) ^†^	0.41 (0.24–1.04)	0.28 (0.15–1.26)	0.54 (0.26–0.89)	0.82
LVEF ^×^	51.31 ± 11.04	55.67 ± 9.26	48.71 ± 10.93	0.003
BMI (kg/m^2^) ^†^	28.7 (27.4–33)	28.4 (27.4–32.4)	30.15 (25.82–33.17)	0.68
Hypertension ^□^	66 (82.5)	38 (84.4)	28 (80)	0.76
Diabetes ^□^	22 (27.5)	14 (31,1)	8 (22.9)	0.45
HbA1c (%) ^×^	7.11 ± 1.47	6.58 ± 1.10	7.67 ± 1.66	0.05
LDL (mg/dl) ^†^	84 (69.8–108)	73(69.8-104)	100.8 (56.6–124)	0.57
Resting HR ^×^	81.9 ± 15.69	84.00 ± 17.25	77.57 ± 12.76	0.05
% peak HR ^×^	77.98 ± 12.25	82.38 ± 11.21	72.31 ± 11.27	0.001
Resting SBP (mmHg) ^×^	127.3 ± 12.65	130 ± 13.39	125.3 ± 11.79	0.1
Resting DBP (mmHg) ^×^	81.5 ± 7.52	80.78 ± 7.305	82.43 ± 7.8	0.33

NLR—neutrophil to lymphocyte ratio, PLR—platelet to lymphocyte ratio, CRP—C-reactive protein, LVEF—left ventricular ejection fraction, BMI—body mass index, LDL—low-density lipoprotein, % HR—percentage of maximal predicted heart rate during test, SBP—systolic blood pressure, DBP—diastolic blood pressure, * Difference between % VO2 max ≤70 and % VO2 max >70. Data are presented as: ^×^ Mean ± SD; ^□^  *n*, %; ^†^ Median (interquartile range).

**Table 2 medicina-58-00814-t002:** Pearson correlation between NLR, PLR and CPET parameters.

Parameters	NLR	PLR	Resting HR	% Peak HR	% Peak WR	% VO2 Max
NLR	1	0.369 *	−0.087	−0.043	−0.104	−0.133
PLR	0.369 *	1	0.207	0.172	0.105	0.249 *
Resting HR	−0.087	0.207	1	0.594 *	−0.053	0.144
% peak HR	−0.043	0.172	0.594 *	1	360 *	0.448 *
% peak WR	−0.104	0.105	−0.053	0.360 *	1	0.705 *
% VO2 max	−0.133	0.249 *	0.144	0.448 *	0.705 *	1

* *p* < 0.05, NLR—neutrophil to lymphocyte ratio, PLR—platelet to lymphocyte ratio, HR—heart rate, % HR—percentage of maximal predicted heart rate during test, % WR—percentage of the predicted value of maximal work rate, % VO2 max—percentage of the predicted value of maximal oxygen uptake.

**Table 3 medicina-58-00814-t003:** Multivariate regression analysis to predict poor functional capacity.

Variables	Odds Ratio	95% Confidence Interval	*p*
Neutrophil count, ×10^3^/μL	1.00	0.999–1.002	0.523
PLR	1.015	1.004–1.027	0.009
LVEF	1.07	1.003–1.141	0.042
% peak HR	1.088	1.029–1.151	0.003

PLR—platelet to lymphocyte ratio, LVEF—left ventricular ejection fraction, % HR—percentage of maximal predicted heart rate during test.

## Data Availability

Data are available from the corresponding author upon request.

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
