# Peer review of "Value of the Neutrophil to Lymphocyte Ratio and Platelet to Lymphocyte Ratio in Predicting CPET Performance in Patients with Stable CAD and Recent Elective PCI"

_medicina, 2022, doi:10.3390/medicina58060814_

Round 1

Reviewer 1 Report

This is an interesting, overall well written and well conducted in-depth study on two easily and readily available potential predictors of systemic inflammation, low grade chronic inflammatory status, and CV disease burden.

Some minor issues should be pointed out:

  1. “However, current literature offers limited data regarding the role of [NLR as] readily available inflammatory biomarkers in predicting exercise performance in CAD patients”: it’s true, but it has been recently published a potential role of NLR in predicting the response to lipid lowering effectiveness in high CV risk subjects (Scicali et al, PMID: 34627693); this significantly enhance the role suggested by the authors.
  2. …”resting BP values greater than 140 and/or 90 mmHg for systolic and/or diastolic BP, respectively”…
  3. “asting glucose ≥126 mg/dl: fasting.
  4. “Standardized transthoracic echocardiography; CPET”: Italics may not be necessary
  5. “Data is presented as mean ± standard deviation (SD), or as median with interquartile range for continuous variable”: authors should detail less generically the way they followed; It seems they presented data only as mean ± standard deviation (SD). Definitely: are the included variables ALL normally-distributed? If yes, please do not mention median/IQR. If not, to present all data as median/IQR could be preferred. Consistently, the description of the distribution analysis should precede the choice of the data presentation (e.g mentioning KS test).
  6. “Samples t Test//Mann-Whitney U test: please be consistent (see above).
  7. “Multivariate logistic regression analysis”: did the authors dichotomize the variables to be included in the model? Please detail.
  8. “calculating Pearson correlation coefficients” (again normal vs non-normal distribution).
  9. Authors aimed to explore the ability of neutrophil to lymphocyte ratio and platelet to lymphocyte ratio in predicting CPET performance, but overall –also due to the way of their findings- a significant deepening was dedicated only to PLR; I think they could little expand the discussion on NLR, also considering the recent literature.

Reviewer 2 Report

I appreciate work done in conducting and writing the manuscript and thank the Editor for giving me an opportunity to review it.

The authors were interested in evaluating use of PLR and NLR in predicting poor FC among CAD with recent elective PCI. Patient data were collected from one hospital and retrospectively analyzed. The authors found that PLR was independently associated with poor FC. The authors concluded that PLR could be cheap and simple alternative test for predicting poor FC in CAD patients with recent elective PCI.

Major comments:

  1. While the authors mentioned the study hypothesis, they did not explicitly state the aim of the study.
  2. I am not sure whether the study design was actually a retrospective cross-sectional study. Based on the description provided, the investigators first identified a cohort and then determined who had poor and preserved FC. It is likely a retrospective case-control study?
  3. While from the context, I was able to sense what were the exposures of interest, it is necessary to explicitly state them in the methods section. What were exposures? How they were defined/measured? When were they measured? It is crucial to provide sufficient information as to decide whether measurement methods were robust. Especially, it is not clear when blood measurements were collected (baseline, mid-way, or at discharge). How did the authors deal with temporal variability in blood measurements?
  4. It is not clear whether the patients attending the Cardiovascular unit of the Clinical Rehabilitation Hospital in Iași were similar to patients visiting other hospitals? What are characteristics of patients who attend the Cardiovascular unit of the Clinical Rehabilitation Hospital in Iași? Please provide additional description of the hospital?
  5. Another aspect in understanding whether findings could be generalizable is how the investigators recruited patients? What sampling technique was used? Did the authors include all admitted patients?
  6. Since patients were enrolled between 2020-2021, have the authors considered the impact of COVID-19 on these patients? Have they accounted for potential negative effect of COVID-19?
  7. Conceptually, the authors considered two exposures (PLR and NLR) in predicting poor FC. However, it is not clear how the authors investigated the relationships. Whether the authors considered potential confounders? As the PLR and NLR measurements could be explained by other factors, it should be clear conceptual descriptions what factors were considered confounders. What criteria did the authors use to determine covariates to include in the multiple regression model?
  8. While the authors briefly mentioned the results of the multivariable regression model for FC with PLR, it was not clear whether a model with NLR was built as well. Given moderate correlation between PLR and NLR it would be inappropriate to include both exposures in one model.
  9. Given the small sample size, the authors would likely be limited to a few covariates. Overadjustment would likely lead to overfitting which might not be useful for generalizability purposes.
  10. Additionally, the authors did not provide the results of the multivariable analysis. While the association with PLR was mentioned very briefly, unadjusted and adjusted analyses should be presented in a table.
  11. ROC curve is a useful tool in determining a best cut-off level for classification purposes. I believe it is more appropriate to utilize ROC curve when the association was established. Given the lack of methodological rigor and robustness in the analysis (small sample size, residual confounding, selection bias), it is too early to perform such analysis, especially in this study.
  12. At the end of the results section, the authors mentioned that they performed the secondary analysis on LVEF. What was the rationale for this analysis? Why was this analytical approach not mentioned in the methods? How is this analysis aligned with the primary research question? Not sure, whether Table 3 helps in answering the research question?
  13. I believe there are several limitations in the study that were not mentioned and acknowledged in the limitations section.
  14. “Higher PLR predicts an impaired functional capacity in patients with stable CAD and recent PCI.” this is overstatement of the study findings. I am not convinced that the presented analysis was robust enough to conclude this way. There are several other health conditions that might explain the findings (residual confounders, risk factors) which were not adjusted and the sample size unlikely allow to do so.

Minor comments:

  1. In the abstract, it states that the authors aimed to evaluate use of NLR and PLR in predicting FC. After reading the whole manuscript, I believe authors were interested in predicting poor FC (since FC was dichotomized).
  2. Misspelling in line 102 “, asting glucose”
  3. Please provide references for HBP and diabetes definitions.
  4. What was the patient age range? Did the authors exclude patients based on their age?
  5. I do not think that the word prevalence is appropriate in this sentence “Age and the prevalence of cardiometabolic comorbidities (obesity, diabetes, HTN, LDL level)…” Since the sample size is small, patients were recruited from one hospital and the sampling technique is unclear, it is too bald to say prevalence.
  6. Figure 1 presents redundant information which was already presented in Table 1.
  7. Not sure how the second and third paragraphs in the Discussion section relevant to the study findings?
